# A survival model for course-course interactions in a Massive Open Online Course platform

**Edwin H. Wintermute**[1]*, **Matthieu Cisel**[2], **Ariel B. Lindner**[1]

**1** Université de Paris, INSERM U1284, Center for Research and Interdisciplinarity (CRI), Paris, France,
**2** Institut des Humanités Numériques, CY Cergy Paris Université, Cergy, France

\* jake.wintermute@cri-paris.org

## Abstract

Massive Open Online Course (MOOC) platforms incorporate large course catalogs from which individual students may register multiple courses. We performed a network-based analysis of student achievement, considering how course-course interactions may positively or negatively affect student success. Our data set included 378,000 users and 1,000,000 unique registration events in France Université Numérique (FUN), a national MOOC platform. We adapt reliability theory to model certificate completion rates with a Weibull survival function, following the intuition that students "survive" in a course for a certain time before stochastically dropping out. Course-course interactions are found to be well described by a single parameter for user engagement that can be estimated from a user's registration profile. User engagement, in turn, correlates with certificate rates in all courses regardless of specific content. The reliability approach is shown to capture several certificate rate patterns that are overlooked by conventional regression models. User engagement emerges as a natural metric for tracking student progress across demographics and over time.

## Introduction

In recent years millions of students have registered for thousands of newly created online courses with topics spanning the range of human knowledge [1]. Massive Open Online Courses (MOOCs) are a subset of online courses defined by a commitment to open access and unlimited registration [2]. MOOC use is increasingly mediated though MOOC platforms, websites that offer centralized access to many courses through a standard user interface. The rising popularity of MOOCs motivates the detailed study of user outcomes.

The large user-bases and digital format of MOOCs generates large data sets in which a variety of studies have sought the keys to user success. Previous work has identified course-specific features of style and content that characterize highly effective MOOCs [3–6]. Other studies look outside the course for student-specific demographic and social factors that affect performance [7–9]. Here we consider the interaction-specific factors that come into play when users register for multiple courses. Although the central organization of MOOC platforms encourages multiple registrations, there has not yet been a systematic study of course-course effects.

**Data Availability Statement:** All files are available from the Zenodo repository (DOI: 10.5281/zenodo.3969240)

**Funding:** This work was supported by the Bettencourt Schueller Foundation through a CRI

Research Fellowship to EHW. The funders had no role in study design, data collection and analysis, decision to publish, or preparation of the manuscript.

**Competing interests:** The authors have declared that no competing interests exist.

Our data is collected from France Université Numérique (FUN), the French national MOOC platform. The FUNCERT data set logs 1,000,000 registrations from 378,000 unique users 140 courses over a 23-month period. Each registration event records a user ID, a time stamp, and whether a certificate was obtained. On average, 8.1% of course registrations produced a certificate.

The framework of this study is a statistical model for the probability that a given registration event will produce a certificate. Central to this model is the effect of registration bursts, in which users register for multiple courses within a short period of time. We present a simple mathematical relationship between burst size and expected certificate rate.

Our model further accounts for user-dependent and course-dependent effects on certificate rate by associating each registration event with terms for user engagement and course difficulty. A user's engagement term is estimated using their complete registration profile and therefore reflects the influence of each registered course on every other.

Tracking difficulty and engagement separately, we follow the progress of users who return to FUN for multiple MOOCs. Returning users are shown to be both more engaged and more inclined to register difficult courses. Similarly, changes in engagement and difficulty combine to drive increases in certificate rates for older users.

Our model is structured as a Weibull survival function, a formal framework commonly used to describe failure rates in mechanical systems with many independent modes of failure. In this way, the certificate rates of MOOC users are connected to the well-developed statistical methods of reliability analysis. This approach is shown to outperform conventional logistic regression in describing key global patterns in certificate rate.

## Methods

### Ethics statement

This study uses only non-interventional and fully anonymized data collected by a 3rd party with user consent. Ethical review was provided by the INSERM Ethics Evaluation Committee.

### Data collection and anonymization

Course registration and user certificate profiles were collected from October 2013 to September 2015 using the Google Analytics platform. Registrants were asked to self-report their gender and birth year. Personally identifying information including course names, user names and email addresses were removed prior to analysis. The privacy policy and terms and conditions of use for the France Université Numérique are available on the platform website (www.fun-mooc.fr).

### Data cleaning and pre-processing

The provided data set included 1,048,566 registration events representing all activity recorded on the platform during the collection period. Of these, 34 could not be assigned to a specific course, user, timestamp, or certificate status and were discarded. Of 140 courses represented in the registration events, 49 did not offer certificates or did not record any awarded certificates during the study period. Registration events for all 140 courses were used to calculate registration burst sizes but only the 91 certificate-awarding courses could be assigned course difficulty terms within the model.

### Characteristics of and Inputs to the $\beta END$ model

The $\beta END$ model was developed as a novel parametric statistical model reflecting our hypothesized mechanisms influencing MOOC certificate rates as articulated in the main text. It

calculates the probability that a registration event will result in a certificate as a function of $\beta$, a Weibull shape parameter, $E_U$, a derived parameter representing user engagement, $N$, the number of simultaneously co-registered courses and $D_C$, a free parameter representing course difficulty. The $E_U$ term for each registration event is calculated as the sum of $D_C$ terms for all courses registered by that user during the study period. A full mathematical description of the $\beta END$ model is provided as S1 File.

### Clustering of registration events into bursts

Registration events were assigned to bursts using agglomerative clustering in the MATLAB Statistics Toolbox with a shortest-distance linkage function. A hierarchical cluster tree was generated using the time between registration events as a distance metric and a nearest-neighbor linkage function. Bursts were separated from the cluster tree using a delay time cutoff of 8 hours.

### Parameter fitting by maximum likelihood

In total we fit 92 free parameters: 91 values of $D_C$ for the 91 certificate-awarding courses and a single value of $\beta$, the Weibull shape parameter. Maximum likelihood estimation of the free parameter values was performed using the fminsearch function of MATLAB. Parameter confidence intervals were determined using 100 bootstrap replicates in which 330,000 users were randomly assigned to a training groups and the remainder to a test group. All reported statistics of the performance of the model are on users of the test group. A complete regression table including best-fit parameter values is provided as S4 Table of S1 File.

### Performance benchmarking with logistic regression

The $\beta END$ model was compared to a standard logistic regression model developed with the same number of free parameters. The logistic regression model predicts the log-odds of course certification as a linear combination of course difficulty, $D_C$, user engagement, $E_U$, and a coefficient weighting the burst size N. Details of the construction of the logistic model are provided as S1 File.

## Results

### Derivation of the $\beta END$ model as Weibull survival function

Fig 1 reviews the structure of the $\beta END$ model. Each course is assigned a difficulty term, $D$, that summarizes all course-specific features contributing to the certificate rate. High difficulty courses may exhibit, for example, advanced subject matter, heavy work-loads, or an unappealing presentation style. Each user is associated with an engagement term, $E_U$. High engagement users may enjoy natural aptitude, prior preparation, or a willingness to invest time.

We further assume that a user's engagement is determined by the set of courses they have chosen to register (Fig 1A). Intuitively, we expect users to seek out courses that match their intrinsic level of skill and motivation. We therefore apply a course engagement term, $E_C$, for each course and calculate $E_U$ as the sum of $E_C$ for all courses registered by a user during the data collection period.

With these estimates of user engagement and course difficulty, we next sought an appropriate mathematical framework to describe the dynamics of user persistence or withdrawal. Student drop-out behavior is recognized as a complex social phenomenon with many causes [10]. A course might be too advanced, too demanding, or too dull. Personal, professional or social circumstances can change [11]. The many factors that keep a student actively working toward

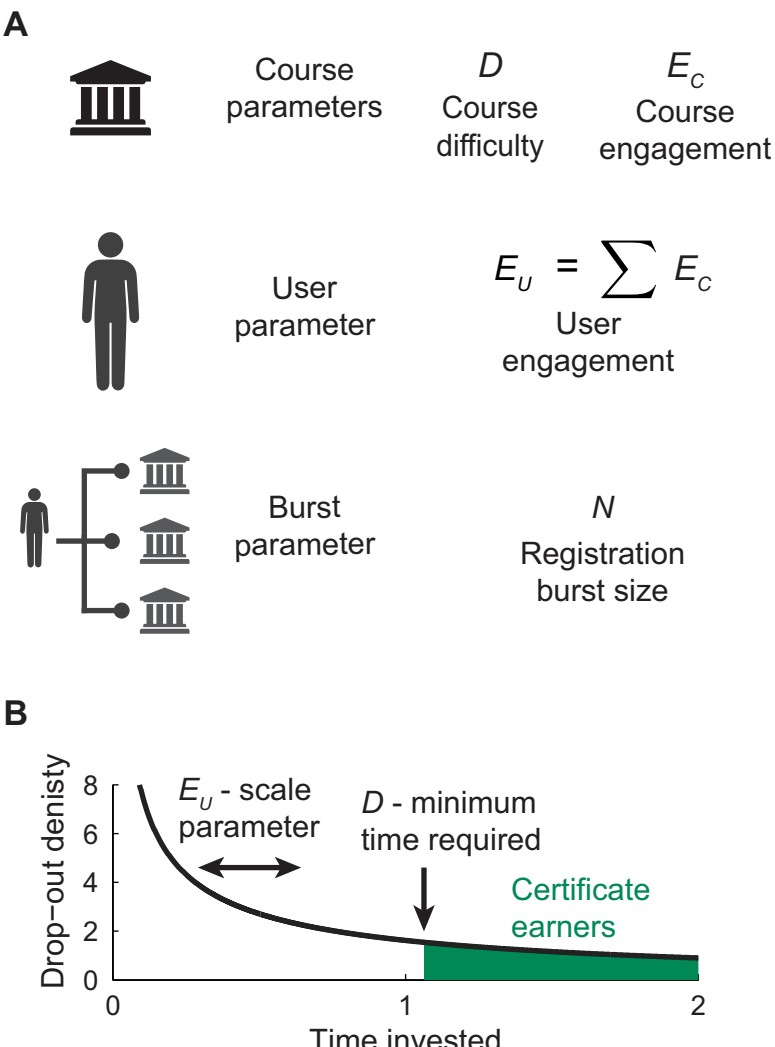

**Fig 1. The βEND model parameterizes a Weibull survival function and accounts for the effect of multiple registrations.** (A) Each registration event is associated with a course difficulty, $D$, and a user engagement $E_U$. User engagement is estimated as the sum of course engagement, $E_C$, for all courses registered by a user during the study period. Users may register for multiple courses in a single burst event, with $N$ the number of registrations. (B) The $E_U$ and $D$ terms have intuitive interpretations in a Weibull probability density function. Each student who registers for a course remains enrolled for a certain time before dropping out. The $D$ term can be considered a minimum time investment with which a student can earn a certificate. The $D$ term is therefore analogous to course difficulty. The $E_U$ term functions as a scale parameter that shifts the distribution toward higher investment times. We therefore interpret $E_U$ as a representation of user engagement.

a certificate can be thought of as links in a chain. A single point of failure is sufficient for drop-out to occur.

The weakest-link concept is formalized by the Weibull survival function. In any system with multiple essential components, the failure time of the system is set by the minimum failure time among the components. The extreme value theory provides that, under appropriate conditions, the distribution of minimum failure times will approach a Weibull [12]. This is true regardless of the model chosen for the failure of the individual components. For this reason, the Weibull distribution is mechanistically appropriate to describe failure rates in many complex systems [13].

The terms $E_U$ and $D$ are therefore used to parameterize the survival function of a Weibull distribution. $D$ becomes the time-to-failure threshold: a user must persist beyond $D$ to obtain a course certificate. $E_U$ serves as the scale parameter for the distribution, with larger values indicating a greater density of long-term survivors. We assign $\beta$ as a Weibull shape parameter. Values of $\beta$ less than 1, as we discover in this data set, indicate a progressively decreasing rate of failure.

$$S = exp\left[-\left(\frac{D}{E_U}\right)^{\beta}\right] \tag{1}$$

Finally, we consider the fact that students who register for many courses within a short time period are forced to divide their efforts. $N$ represents the number of courses registered during a single registration burst event, defined below. We make the simple assumption that probability density is divided evenly among multiple registrations. This leads to the expression $C = S/N$, where $C$ is the expected probability of obtaining a certificate. A graphical interpretation of the Weibull survival function is offered in Fig 1B.

The $\beta END$ model seeks to describe the effect of multiple registrations on certificate rates. However, in the FUN data set, 56% of users register for only one course. For these users, $E_U = E_C$ and Eq 1 rearranges to:

$$E_C = \frac{D}{-log(S)^{\frac{1}{\beta}}} \tag{2}$$

We used this relation to constrain $E_C$ as a function of $D$ for each course. Because our model incorporates the observed baseline certificate rate for single-registered users of each course, the predictive power is limited the changes in certificate rate expected from multiple registrations.

## Registration events can be clustered into well-defined bursts

When users register for multiple courses simultaneously their attention is divided and their chance to earn a certificate in each course is likely to decrease. Our model quantifies this effect with the parameter N, registration burst size, and predicts that certificate rates will decrease precisely as 1/N. We first sought a method to rigorously define N for each registration event.

The histogram of same-user registration delay times revealed a bimodal distribution (Fig 2A). Following a registration event, the chance that the same user will register for another course reaches a local minimum after a delay of around 8 hours, with most re-registrations occurring after either significantly shorter or longer delay times. We therefore assigned registration events to bursts using agglomerative clustering with a time threshold of 8 hours. Varying this threshold between 4–24 hours changed the total number of identified clusters by less than 1% (S3 Fig of S1 File).

## Burst size and burst number show opposing effects on certificate rates

Fig 2B shows the importance of accounting for burst structure when considering the effect of multiple registrations on certificate rates. Globally we observed a modest positive relationship between a user's total registration count and their chance of earning a certificate in any particular course. However, this effect could be decomposed into a stronger positive impact of total burst events and a significant negative impact of within-burst registrations. These results are consistent with the idea that users divide their attention among multiple co-registered courses, resulting in a lower per-course certificate rate. On the other hand, returning to the platform

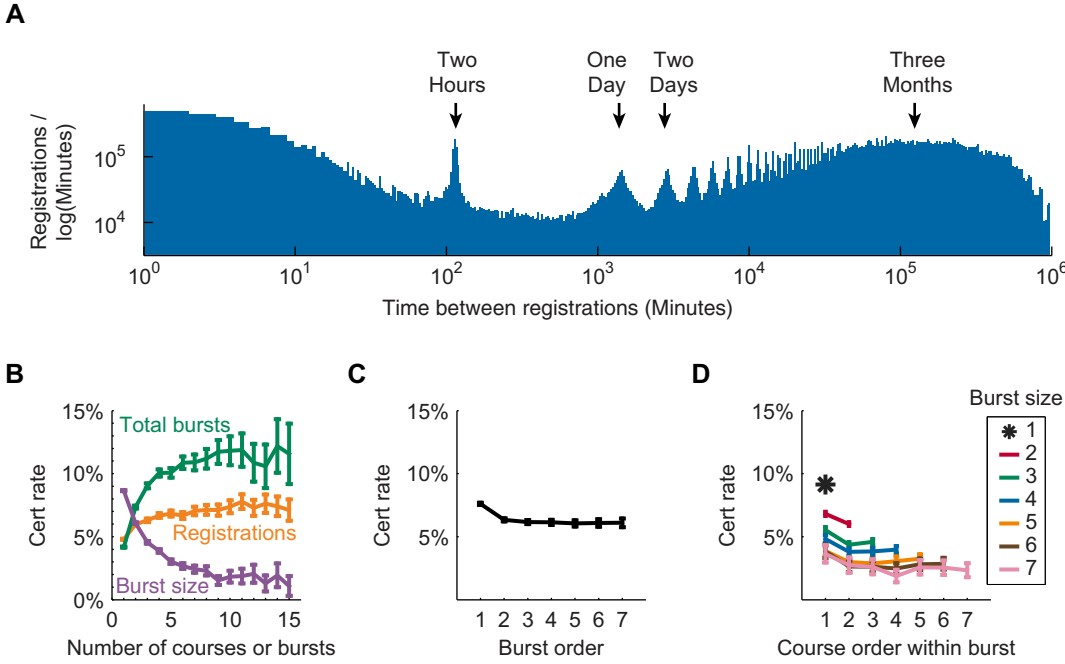

**Fig 2. Bursting dynamics in registration events affect certificate rates.** (A) A histogram showing delay times between same-user registration events presented as log-scaled event frequency binned by log-scaled time. The peak at two hours may correspond to confirmation message sent to new user accounts. The day-night cycle of user activity can be seen in peaks with a period of 24 hours. A clustering threshold of 8 hours was used to group registration events into bursts. (B) User per-course certificate rates as a function of total registrations, total burst events or burst size. Larger bursts were associated with lower per-course certificate rates while more bursts indicated higher rates. (C) Per-course certificate rates for registration bursts according to the time order in which they were registered. A user's first registration burst is associated with a higher certificate rate. No significant changes are observed across subsequent bursts. (D) Certificate rates for courses within a given burst according to the time order of registration. Larger bursts were associated with globally lower per-course certificate rates. The first course registered within a burst is more likely to be certified with no change across later bursts. Error bars for BCD are Bernoulli standard error.

after a significant time delay does not appear to cause distraction and may instead be an indication of greater motivation for online learning.

As well as being clustered in time, registration events are ordered in time. Fig 2C and 2D presents the order-dependent structure of certificate rates both within and between burst events. Per-course certificate rates were highest in a user's first registration burst then stable across second and subsequent bursts. A similar pattern was observed within a burst, where the first-registered course is more likely to be certified but no further order effects are apparent. First registrations may be associated with higher user interest or motivation because they were the original inspiration for the registration activity. However, because the effect of registration order was small relative to the effect of burst size, we did not attempt to account for it within the framework of our simplified model.

## Co-registration affects expected certificate rates

Fig 3 compares certificate rates derived from the *βEND* model with rates in the FUN data set. Registration events were assigned certificate probabilities over two orders of magnitude that agreed well with empirically observed rates. In contrast, a conventional logistic regression model systematically overestimated certificate probabilities for low-probability events (Fig 3A).

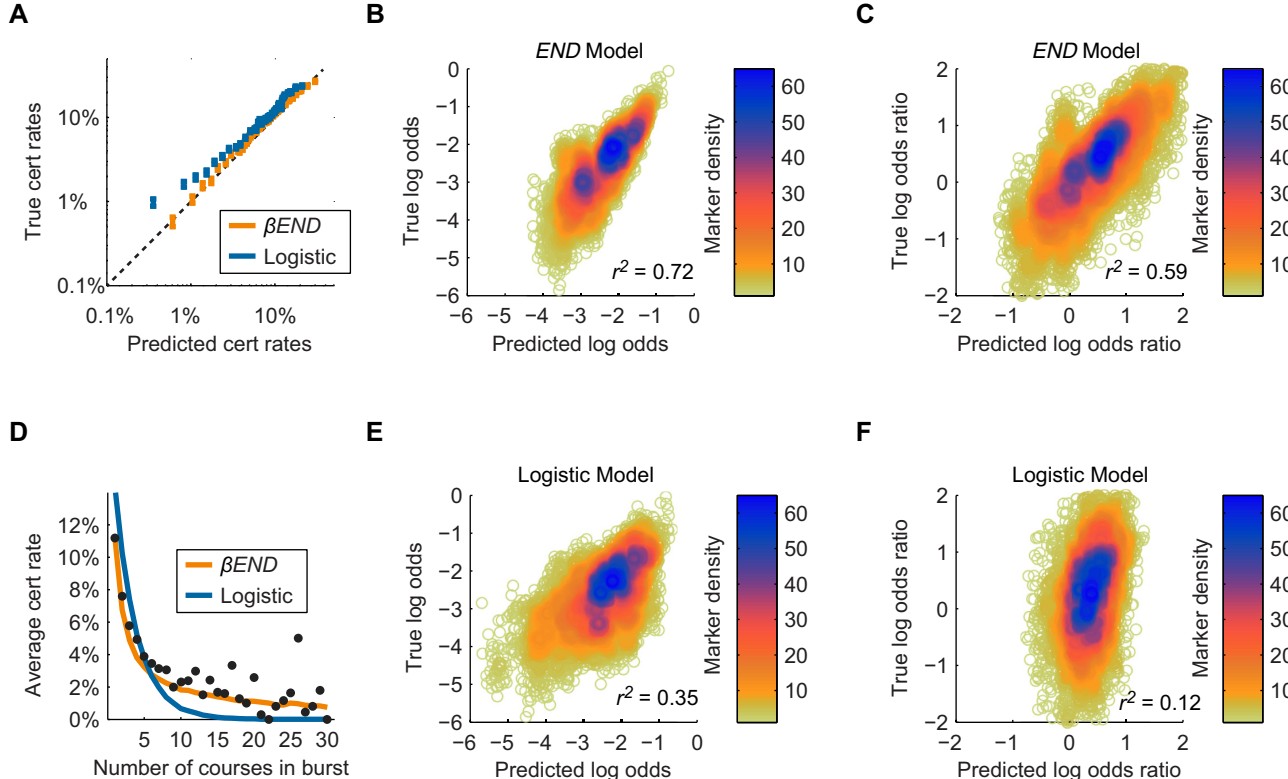

**Fig 3. The βEND model outperforms logistic regression in predicting key features of the FUN data set.** (A) A Hosmer-Lemeshow plot comparing certificate rate predictions of the βEND and logistic models. All recorded registration events were sorted into 20 bins by predicted certificate rate. The mean predicted and true observed certificate rate for each bin is plotted for both models. While the βEND model predictions show a linear relation over two orders of magnitude, the logistic model systematically overestimates certificate rates for low-rate registration events. (B) Users were grouped by co-registration cohort, defined as the set of users who registered for each possible set of two courses. The βEND model was used predict the log odds of obtaining a certificate in the first course conditioned on co-registration in the second. Predicted and observed log odds values were linearly correlated (r2 = 0.72). (C) Log odds ratios for each co-registration cohort were calculated as the log odds of obtaining a certificate given co-registration minus the log certificate odds for users who registered for only the course alone. Log odds ratios predicted by the βEND model correlated with observed values (r2 = 0.59). (D) Registration events were grouped by burst size, the number of registrations recorded by a user within a short time period. The approximately inverse relationship between certificate rate and burst size was well described by the BEND model but not by conventional logistic regression. (E) Predictions obtained through logistic regression for the certificate log odds of co-registration cohorts were less well correlated to observed log odds (r2 = 0.35). (F) Similar predictions obtained by logistic regression for certificate log odds ratios relative to single-registered users correlated with observed values (r2 = 0.12).

We define a co-registration cohort as an ordered pair of two courses and the set of users who have registered for both. Users may also have registered for additional courses. The conditional certification rate for the cohort is the certificate rate for the first course among users who also registered for the second. Of 8190 possible co-registration cohorts, 7321 produced at least 5 certificates and were included in the analysis.

The certificate rates of each co-registration cohort were well described by the *βEND* model (Fig 3B and 3C). Predicted certificate log-odds correlated with observed values (Pearson's $r^2$ = 0.72). We also examined the change in certificate rates for each course associated with co-registration in each other course, expressed as a log odds ratio (Fig 3C). Empirically observed log odds ratios correlated with model-derived values ($r^2$ = 0.59).

Fig 3D shows certificate rates as a function of registration burst size. The per-course certificate rate drops in inverse proportion to the number of simultaneously co-registered courses. The effects of burst size on certificate rate were captured by the *βEND* model but overestimated

by logistic regression. The logistic regression model also performed relatively poorly in predicting the certificate log-odds (Fig 3E and 3F).

## Engagement governs the effect of co-registration

Eq 1 can be log transformed twice to isolate the respective contributions of difficulty and engagement to the certificate rate.

$$log(-log(S)) = \beta \cdot log(D) - \beta \cdot log(E) \qquad (3)$$

We made use of this linearization to express the effect of co-registration as the loglog transformed certificate rate of each co-registration cohort minus the loglog transformed rate of users who registered for only a single course. Because both sets of registration events share the same course difficulty, the result reflects the difference in user engagement between the single-registered and the co-registered populations.

Fig 4A compares the certificate rate of users who registered for only a single course with that of every possible co-registration cohort. 32% of courses saw significant increases in certificate rate given co-registration while 35% saw decreases. These changes were homogenous across the set of possible co-registrations. In other words, courses that benefited from co-registration tended to benefit from co-registration with any other course.

The global effect of co-registration on double-log transformed certificate rates was largely governed by log user engagement (Fig 4B). In courses with below average user engagement, co-registration will generally increase expected engagement levels. In high-engagement courses, co-registration will generally decrease expected engagement and therefore certificate rates.

## Engagement levels vary with time and age

The *βEND* model assigns user engagement and course difficulty scores to each registration event. In this way, it allows user-specific and course-specific factors affecting user registration to be decoupled and separately related to other relevant data.

We first looked at the changes in model-derived parameters for users who returned to the platform multiple times during the data collection period (Fig 5A, 5C and 5E). Two or more independent registration burst events were recorded for 30% of users. Certificate rates did not change significantly following re-registration (Fig 5A). However, both course difficulty and course-associated engagement levels were found to significantly increase (Fig 5C and 5E).

The FUN data set also includes a self-reported age for 93% of users. Certificate rates increase consistently with user age (Fig 5B). This increase is associated with a decline in course difficulty between the ages of 20 and 50 (Fig 5D). Following age 55 we observed a significant increase in the engagement term (Fig 5F). Therefore, the positive trend in certificate rates is driven first by course difficulty, then by user engagement at later ages.

## Discussion

Recent debates have focused on the causes of putatively low MOOC certificate rates [14]. We found that a significant drop in average certificate rate is linked to bursting registration behaviors. Bursting registrations are likely to be common on other MOOC platforms because activity bursts are a general feature of internet user behavior [15].

This result suggests that certificate rates could be increased by constraining burst sizes, for example by limiting users to one new registration per day. However, it is not clear that users would be well served by this. Registration bursts might represent a kind of course-shopping strategy through which users optimize their course selections [16]. As evidence of this, we

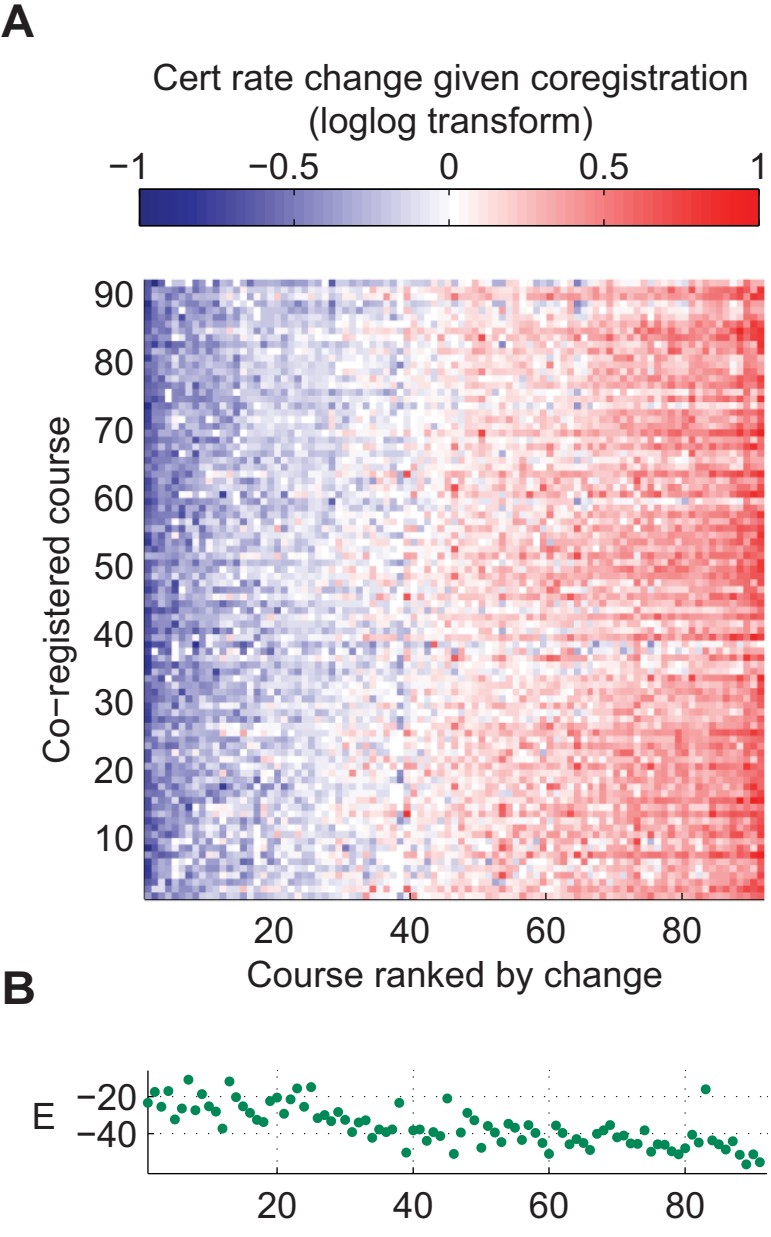

**Fig 4. Course-course interactions are uniform with respect to the co-registered course and are governed by user engagement.** (A) Heatmap showing the effect of co-registration on certificate rates. The effect is expressed as the loglog transformed certificate rate of a given course, conditional on co-registration with each other course, minus the loglog transformed certificate rate of users who registered for only each course alone. The registered courses indicated on the horizontal axis were ordered by the median effect of co-registration. (B) Model-derived course engagement values for each course, with courses ordered as above. Courses with low engagement courses benefit the most from co-registration, regardless of which other course was co-registered.

observed that users who register in larger bursts earn more total certificates, even as their average certificate rate declines.

If many burst registrations are never invested with serious user effort, then the average certificate rate will not reflect a typical user's experience. Instead, we propose certificates-per-burst as a simple metric for the expected overall success of each user upon each approach to

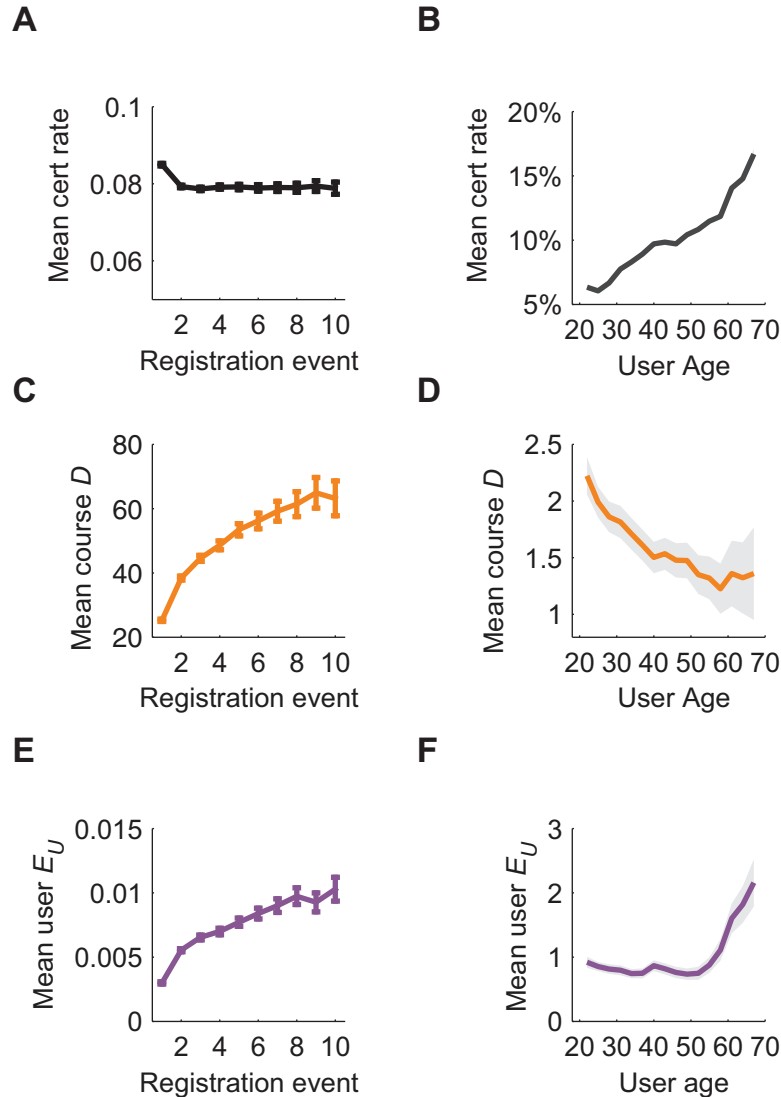

**Fig 5. Model-derived difficulty and engagement values provide mechanistic insight into trends in user certificate rates.** (A,C,E) Users who return to the platform for multiple registration bursts do not show significant increases in certificate rate. The βEND model indicates that both user engagement and course difficulty consistently increase with re-registration. Increases in user engagement scores may reflect positive learning outcomes. Error bars represent Bernoulli standard error. (B,D,F) Certificate rates increase significantly and monotonically with user age. The βEND model suggests this pattern is driven by decreasing course difficulty from younger to middle aged users, then by increasing engagement among older users. Shaded regions indicate 95% confidence intervals.

the platform. The per-burst certificate rate for FUN is 12.2%, 1.5-fold higher than the raw certificate rate.

Survival models may help to reveal the causes and consequences of the low-engagement registrations that frustrate many MOOC providers [17]. One possibility is that a certain proportion of registered users have a strictly zero probability of obtaining a certificate. Cure models in survival analysis [18] or zero-inflated models in organizational research [19] are appropriate for describing such a fully disinterested population. The *βEND* model, in contrast, allows user engagement levels to be low but non-zero and suggests that small increases in engagement will increase raise rates among low-engagement users. Future studies could

resolve these competing hypotheses by identifying user behaviors consistent with strictly zero or non-zero engagement. Such studies will require high temporal resolution in the period immediately following registration, when survival models indicate a fast transition from persistence to drop-out.

The global dynamics of drop-out in the *βEND* model are summarized by the Weibull shape parameter, *β*, which took a best-fit value of 0.13. While shape parameter greater than one is consistent with conventional aging and increasing frailty, shape values less than one indicate that survivors become more reliable over time. The MOOC user population could be said to experience burn-in, with users who persist in a course becoming increasingly inclined to finish it. Other work has shown that investments of user time and effort produce non-linear increases in certificate rate [5, 7, 20].

The *βEND* model outperformed logistic regression in fitting this data set. The symmetrical logit link function is known to exhibit bias when applied to asymmetrical data sets where the chance of success approaches zero much faster than one [21]. Previous work has shown generalized Weibull linkage functions to outperform logistic approaches in describing mechanical failure, mortality, and other systems characterized by weakest-link scaling [22].

The success of the *βEND* model demonstrates the information richness of complete user registration profiles. However, similar data sets may be difficult to obtain for other user populations. While FUN is the predominant French-language MOOC platform, several platforms compete for the attention of English-speaking students. Even within a single platform, user data may be segregated among participating institutions, constraining analyses to the institution level [20]. Any comprehensive MOOC performance model should thoroughly account for co-registration, which may occur across platforms.

We found co-registration was associated with significant changes in certificate rate. Out of 91 courses, 27 saw a 2-fold increase or more in certificate rate for users that had co-registered another course. Another 31 courses saw a 2-fold decrease in rates given co-registration. Remarkably, the certificate rate changes were broadly similar regardless of which other course was co-registered.

The effects of co-registration were mediated by course engagement levels. Our model associates each course with an engagement score and estimates a user's engagement as the sum of their registered courses'. In courses with below-average engagement, co-registration tends to raise a user's expected engagement level. The opposite is true for courses with above-average user engagement, where co-registration is associated with lower certificate rates.

Users who returned to the FUN platform to take new courses were not more likely to earn certificates, even after multiple rounds of re-registration. This is surprising given the expectation that users become more knowledgeable and skilled through coursework. The model-guided analysis revealed that returning users select courses of progressively higher engagement scores and higher difficulty, two opposing trends that result in no net change in certificate rate.

This result underscores that certificate rates alone should not be used to track user progress because they also reflect the difficulty of the selected courses, which may systematically vary between groups and over time. The engagement term of the *βEND* model effectively controls for course difficulty and reports only user-dependent contributions to success. In the case of returning users, increasing engagement could reflect progressive learning acquired through coursework. Alternately, it may indicate a tendency of users who are a priori more engaged to concentrate in specific courses as they become familiar with the platform.

Decoupling course difficulty from user engagement provides insight into other demographic trends associated with user success. Overall certificate rates increase monotonically with user age. Between youth and middle age, the trend is driven by decreasing course

difficulty. This could mean that middle-aged users value certificates more highly and select courses where they are more likely to obtain one. Alternately, it could indicate that courses appealing to middle-aged professionals are less technical or challenging than courses selected by college-aged MOOC users. The ages of 45–65 are marked by a progressive increase in user engagement. This might imply more time, greater ability, or simply more enthusiasm for digital learning among older users.

A complete understanding of the factors that promote effective web-based learning will require major new programs in education research [23]. Our model quantifies user engagement and course difficulty but does not describe how those properties manifest in the real world. Does user engagement reflect intelligence, motivation, free time or something else? What makes one course difficult and another easy? Interdisciplinary efforts are called for to connect model-derived statistical insights with detailed descriptions of user psychology and course design. Such work may reveal strategies to enhance user engagement and reduce course difficulty.

Many of the concepts currently being used to describe digital experiences were developed for the pre-digital era of distance education [24–26]. MOOC data sets are not only much larger, but document new kinds of behaviors and relationships that cannot be easily described with established theories [27]. New theoretical frameworks are needed to describe the fluid, networked experience of taking an online course online and generate actionable models for making those courses better.

The $\beta END$ model formally connects MOOC certificate rates to systems reliability theory. Within this domain a rich set of mathematical tools will support more detailed analysis of the dynamics of dropout. We have demonstrated a parametric approach featuring several strong theoretical assumptions. First, we propose that the mechanisms of dropout follow weakest-link scaling and therefore adopt a Weibull distribution in time. Second, we consider certificate rate probabilities to be divided equally among multiple registrations. While this approximation agrees with our data it is likely that users are making motivated and non-random decisions about how to focus their attention. Finally we assume the independence of user engagement and course difficulty when these two parameters likely feature complex interactions and covariates. As data sets become richer and more time-resolved it will become possible to develop parametric or non-parametric reliability models that are richer or more generalizeable. Future work will also need to consider MOOC formats that continuously evolve to meet the needs of users and providers [17].

Reliability engineering is the application of survival analysis to identify model-guided interventions that reduce failure rates in complex systems. Efforts to improve MOOC user outcomes may benefit from the quantitative framework applied to other systems characterized by stress, fatigue and random break-downs.

## Supporting information

**S1 File.**
(DOCX)

## Acknowledgments

We thank Catherine Mongenet and the FUN platform for providing access to the anonymized MOOC registration data. Dusan Misevic and Ignacio Atal provided feedback on the structure and content of this manuscript. Marc Santolini and Liubov Tupikina provided advice on analyzing and visualizing co-registration data.

## Author Contributions

**Conceptualization:** Edwin H. Wintermute, Ariel B. Lindner.

**Data curation:** Matthieu Cisel.

**Formal analysis:** Edwin H. Wintermute.

**Funding acquisition:** Edwin H. Wintermute, Ariel B. Lindner.

**Methodology:** Edwin H. Wintermute, Matthieu Cisel.

**Writing – original draft:** Edwin H. Wintermute.

**Writing – review & editing:** Edwin H. Wintermute, Matthieu Cisel, Ariel B. Lindner.

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
