## [Decision Letter · Decision Letter 0]

13 Mar 2020

PONE-D-19-22015

A survival model for course-course interactions in a Massive Open Online Course platform

PLOS ONE

Dear Dr. Wintermute,

Thank you for submitting your manuscript to PLOS ONE. After careful consideration, we feel that it has merit but does not fully meet PLOS ONE’s publication criteria as it currently stands. Therefore, we invite you to submit a revised version of the manuscript that addresses the points raised during the review process.

ACADEMIC EDITOR: Please attend to each point raised by the two reviewers, and improve the presentation of the ideas for more clarity.

We would appreciate receiving your revised manuscript by Apr 27 2020 11:59PM. To enhance the reproducibility of your results, we recommend that if applicable you deposit your laboratory protocols in protocols.io, where a protocol can be assigned its own identifier (DOI) such that it can be cited independently in the future. For instructions see: http://journals.plos.org/plosone/s/submission-guidelines#loc-laboratory-protocols

We look forward to receiving your revised manuscript.

Kind regards,

Mingming Zhou, Ph.D.

Academic Editor

PLOS ONE

Journal Requirements:

Reviewers' comments:

Reviewer's Responses to Questions

**Comments to the Author**

1. Is the manuscript technically sound, and do the data support the conclusions?

Reviewer #1: Yes

Reviewer #2: Yes

2. Has the statistical analysis been performed appropriately and rigorously? 

Reviewer #1: I Don't Know

Reviewer #2: I Don't Know

3. Have the authors made all data underlying the findings in their manuscript fully available?

Reviewer #1: No

Reviewer #2: No

4. Is the manuscript presented in an intelligible fashion and written in standard English?

Reviewer #1: No

Reviewer #2: Yes

5. Review Comments to the Author

Reviewer #1: Note on data availability – the link provided for the data goes to the general home page of the FUN MOOC platform; there is no indication of how to access the specific data set.

Note on research ethics – the study uses data from human participants, so it is unclear why the ethics statement field is marked N/A

This paper presents some interesting and novel findings relating to the likelihood of MOOC completion, taking into account the effect of users’ overlapping registrations on multiple courses. The suggestion to measure success per burst (of multiple registrations) rather than success per course could be of particular interest to researchers interested in MOOC success factors. In addition the novel approach to analysing data in this domain highlights some interesting characteristics which have not been evident from previous work, for instance the dichotomy between some courses where users tend to do better in the context of multiple registrations and others where the opposite occurs. However, I found the presentation of this paper very confusing and therefore difficult to follow. The paper follows an unconventional ordering, with method reported at the end of the paper. I would normally expect to see the following order: introduction, method (data collection, analysis method), results and discussion. In addition I feel that the decisions regarding which material to include under supplementary material and which to foreground in the paper could be reconsidered. Many potentially interesting findings are included under supplementary material, without clear signposting in the main results section, and then appear summarised in the discussion without an indication of where they have come from. The results reported in the main paper focus mainly on the extent to which the model fits the data, but this is also confusing with much of detail on parameters included in the supplementary material and the meaning of some of findings left quite unclear (e.g. what does it mean in figure 2A to report certification rates at 10-2?). In general the relationship between model parameters and measurable variables could be made much clearer; for example is user engagement something you can measure in reality or purely a theoretical construct here. Also the role of the model in the work could be clarified further by explaining where novel findings about behaviour only become observable by using the model. For me, the evidence regarding the fit of the model compared to regression is less interesting than the novel understanding of behavioural patterns that emerge from the analysis (and form the focus of much of the discussion) and may be more appropriate for the supplementary section.

Reviewer #2: Thank you for the opportunity to review your paper. I think it's definitely an enjoyable read and an interesting topic you are addressing.

The paper is understandably written and original. Therefore, I would like to address certain keypoints where I think this manuscript could be improved before publication. These keypoints are thoughts and tips for analyzing the data in a different way, which might lead to somewhat different conclusions. I would like to see these points addressed or answered.

The first two keypoints are from a rather 'methodological point-of-view', considering my statistical background and my own work in survival modeling.

(1) You mention that only 8.1% of all registrations lead to a certificate. I think this number of events is very low, especially when facing such a big dataset. Therefore, I would like you to consider fitting a cure model. This class of survival models takes into account that a proportion of your sample is 'not susceptible' for the event-of-interest (i.e., certification). I guess that some courses might be way too difficult, so barely anyone will get a certification OR some users (cfr. engagement) are roamers who subscribe without any intention of completing a course (e.g., for revision or whatever intent), so they will never complete a course. A third possibility is maybe the combination too (hardest courses can only be completed when the user is sufficiently engaged).

I think these analysis tools might yield much more robust results. Anyway, I would like to hear your thoughts on it and address this issue/possibility in the paper.

(2) It remains unclear to me, how you addressed the course-course interaction in the paper. It is not entirely clear which model you used (a fixed effects model with course dummies I suppose). This made it hard to understand how you link your results to course-interactions.

I would, again, like to propose an alternative: use a cross-level frailty model. Your course-user observations can be seen as having a cross-level structure which you can take into account by considering your observation to be nested in both users and courses. Hence, you get more specific estimates on user-level effects (User Engagement) and course-level effects (Difficulty). These models immediately allow to infer on the correlation structure of random effects.

(3) Why do you need cross-validation? Why is this used on your data and what are the results? Did you use it to assess standard errors? To train the model in a Machine-Learning-like way? I don't get why this is necessary for your model?

Next, I would like to address certain keypoints which will make the paper easier to understand for a broader audience.

(4) Could you please elaborate a little on the BEND model you are using. I had never heard about this model and, correct me if I am wrong, I think not many other researchers will recognize the model immediately. I think I would be great to know a little about the model and its relation to more common survival models, possibly even non-parametric survival models such as the Cox model.

In this, I would also like to see a short discussion on how well the model fits. It has been found multiple times that misspecifying a baseline hazard might lead to biased results (which could be avoided by juxtaposing your model with a Cox model, which leaves the baseline hazard unspecified).

(5) I would like a regression table equivalent to judge which covariates are in your model. It is not clear to me whether the discussion of the results comes from one integrated model (where you display marginal effects) or a series of split-sample (post-hoc) models. Personally, I think the second option may suffer from some severe drawbacks (e.g., omitted variable bias, confounding effects, ...).

(6) What is the rationale of the registration bursts? Why don't you left-truncate the data, so every observation line is a registration spell?

You can take the number of simultaneous spells into account as a covariate (or allow differences in baseline?).

Lastly, I will make some minor comments:

- please add a reference to your claim that extreme value theory regards the minimum failure time of n components to be on average Weibull distributed (line 74)

- please check 'weird' line breaks (e.g., lines 98 and 125), it looks like text has disappeared there.

Hence, I will conclude my review of your paper.

I hope to have incited the authors to think about alternative analyses and their potential benefits to the data at hand.

I hope you will engage in a conversation and review of your article along these lines.

I have made myself known, so please feel free to contact me if at any point I was not clear.

6. PLOS authors have the option to publish the peer review history of their article (what does this mean?). If published, this will include your full peer review and any attached files.

Reviewer #1: No

Reviewer #2: Yes: Hans Tierens

---

## [Author Response · Author response to Decision Letter 0]

27 Aug 2020

Individual reviewer comments are addressed in the "Response to Reviewers" attachment.

---

## [Decision Letter · Decision Letter 1]

7 Oct 2020

PONE-D-19-22015R1

A survival model for course-course interactions in a Massive Open Online Course platform

PLOS ONE

Dear Dr. Wintermute,

Thank you for submitting your manuscript to PLOS ONE. After careful consideration, we feel that it has merit but does not fully meet PLOS ONE’s publication criteria as it currently stands. Therefore, we invite you to submit a revised version of the manuscript that addresses the points raised during the review process.

We look forward to receiving your revised manuscript.

Kind regards,

Mingming Zhou, Ph.D.

Academic Editor

PLOS ONE

Reviewers' comments:

Reviewer's Responses to Questions

**Comments to the Author**

1. If the authors have adequately addressed your comments raised in a previous round of review and you feel that this manuscript is now acceptable for publication, you may indicate that here to bypass the “Comments to the Author” section, enter your conflict of interest statement in the “Confidential to Editor” section, and submit your "Accept" recommendation.

Reviewer #1: All comments have been addressed

Reviewer #2: (No Response)

2. Is the manuscript technically sound, and do the data support the conclusions?

Reviewer #1: Yes

Reviewer #2: Yes

3. Has the statistical analysis been performed appropriately and rigorously? 

Reviewer #1: Yes

Reviewer #2: Yes

4. Have the authors made all data underlying the findings in their manuscript fully available?

Reviewer #1: Yes

Reviewer #2: Yes

5. Is the manuscript presented in an intelligible fashion and written in standard English?

Reviewer #1: Yes

Reviewer #2: Yes

6. Review Comments to the Author

Reviewer #1: The paper reads much more clearly now and is now more accessible for educational researchers. Issues around data access and research ethics have also been satisfactorily addressed in the revisions.

Reviewer #2: At the end of August 2020, I had a meeting with the corresponding author of this paper. It was a nice and friendly meeting in which he presented his revised manuscript. We had a friendly discussion about how I saw this paper evolve and its publication potential. I will summarize some of the key take-aways for myself and the authors.

After the data has been made available (to us as reviewers), it became clear to me that most of my previous comments, remarks and suggestions were not applicable to the data at hand. Since there is no time indicator on when the certificate had been obtained and when participants ‘dropped out’, but only on when the participant subscribed for the course, it is impossible to rely on established survival analysis techniques. Without a time stamp on the ‘event of interest’, I agree that there is no clear way to fit any survival analyses to the data (which of course includes their more advanced and specific extensions: frailty models and cure rate methodologies).

This being said, I will now stop devoting much more ‘reviewer space’ to the issue of survival analysis. However, I still do have a few more remarks and suggestions to improve this manuscript before (potential) publication.

Firstly, I think that the authors have formulated some nice and clear answers to the reviewer comments. However, I think that much of these explanations deserve a place in the actual manuscript. More specifically, I am referring to:

1) the elaboration on ‘cured participants’ (i.e., participants who did never subscribe for courses with the aim of obtaining a certificate, and are, thus, insusceptible for the event of obtaining one). I think that this would greatly improve on ways forward in this research domain, as well as a clear guideline on what data is required to continue along this line.

(On a side note, I don’t think that ‘cure’ can be reliably inferred when looking at certification rates across course difficulty (DC). Course difficulty is inferred by the model itself and does not (always) reflect a real time basis. I would suggest restricting Try to make this real time.

Besides this comment, I think that ‘small cure proportions’ is no valid argument to dismiss the reliability of a cure model. In some respect, cure rate models without time varying covariates are really alike a zero-inflated Poisson model and, basically, any selection model or model with truncated dependent variables… In this sense, ‘cure’ proportions of “only” 7% were already found to improve statistical inference (see Blevins et al. 2015 in Organizational Research Methods and Antonakis et al. 2014 in Leadership Quarterly).

2) You claim that the βEND model is “theory rich”, to which I fully agree. However, please bring this more forward in the manuscript. Preferrably, when explaining all the different theory-driven building blocks of the model. I believe that this evidence of a theory-rich model the current manuscript, referring to the added paragraph, is a little short and does not delve too deeply into the ‘theory’ as compared to your response to the reviewers.

In the same light, a theory-rich model necessitates a part of the discussion on how each of your theory-driven choices in developing the building blocks could be ‘untrue’ and how researchers can use their data (which was currently unavailable in your dataset) to improve and test some of these theoretical assumptions/boundaries of the model (for example, baseline hazard, functional form, assumed distributions) as well as elaborate and extend your model using additional data (e.g., course difficulty and user engagement might not be ‘just functions of each other’ but may depend on a set of covariates).

3) I have one additional comment on the clustering process to group subscriptions into bursts. You claim that you wanted to make the model as scalable as possible, which means that researchers in the field can (easily) use your model to gain more insights in their data. However, in what way and to what extent is the clustering process scalable? I think that the rationale of ‘bursts are often found in online user behavior’ is hardly an argument for scalability. If anything, please reflect on this issue in the discussion section of your manuscript.

4) I still don’t think that cross-validation is necessary in this paper and I fear it takes away focus from the core (contribution) of this model. If it is added because of ‘something reviewers will ask for’, I would like to clearly state its purpose in the manuscript and how it may help researchers in their inference.

Lastly, I would like to reinforce a comment of my colleague reviewer (reviewer 1) and counter a response of the authors. I really think that it is advisable to rework the ordering of the sections in the manuscript. I agree that it might be usual in your field to let the methods follow the discussion, but I think for a big portion of the audience of PLOS ONE this ordering seems weird and highly unconventional. I think that readers with a limited statistical background will just skip formulaic expressions and read through them searching for some theoretical foundations of the building blocks of the model. It makes a non-statistical reader also more aware of how the model works, before you present the results of your study. Following the more conventional order (in our field at least); literature, method, results, discussion, readers will better understand the results, where they came from and how to interpret them. I would suggest to write more for the target audience of your journal.

7. PLOS authors have the option to publish the peer review history of their article (what does this mean?). If published, this will include your full peer review and any attached files.

Reviewer #1: No

Reviewer #2: **Yes: **Hans Tierens

---

## [Decision Letter · Decision Letter 2]

7 Jan 2021

A survival model for course-course interactions in a Massive Open Online Course platform

PONE-D-19-22015R2

Dear Dr. Wintermute,

We’re pleased to inform you that your manuscript has been judged scientifically suitable for publication and will be formally accepted for publication once it meets all outstanding technical requirements.

Kind regards,

Mingming Zhou, Ph.D.

Academic Editor

PLOS ONE

Additional Editor Comments (optional):

Reviewers' comments:

Reviewer's Responses to Questions

**Comments to the Author**

1. If the authors have adequately addressed your comments raised in a previous round of review and you feel that this manuscript is now acceptable for publication, you may indicate that here to bypass the “Comments to the Author” section, enter your conflict of interest statement in the “Confidential to Editor” section, and submit your "Accept" recommendation.

Reviewer #2: All comments have been addressed

2. Is the manuscript technically sound, and do the data support the conclusions?

Reviewer #2: Yes

3. Has the statistical analysis been performed appropriately and rigorously? 

Reviewer #2: Yes

4. Have the authors made all data underlying the findings in their manuscript fully available?

Reviewer #2: Yes

5. Is the manuscript presented in an intelligible fashion and written in standard English?

Reviewer #2: Yes

6. Review Comments to the Author

Reviewer #2: I thank the authors for reworking the manuscript for another time.

I am happy that our feedback has been taken to heart and the necessary additions and adaptations are being carried out.

7. PLOS authors have the option to publish the peer review history of their article (what does this mean?). If published, this will include your full peer review and any attached files.

Reviewer #2: **Yes: **Hans Tierens

---

## [Editor Report · Acceptance letter]

12 Jan 2021

PONE-D-19-22015R2 

A survival model for course-course interactions in a Massive Open Online Course platform 

Dear Dr. Wintermute:

I'm pleased to inform you that your manuscript has been deemed suitable for publication in PLOS ONE. Congratulations! Your manuscript is now with our production department. 

Kind regards, 

on behalf of

Dr. Mingming Zhou 

Academic Editor

PLOS ONE